# Online-Dynamic-Clustering-Based Soft Sensor for Industrial Semi-Supervised Data Streams

**DOI:** 10.3390/s23031520

**Published:** 2023-01-30

**Authors:** Yuechen Wang, Huaiping Jin, Xiangguang Chen, Bin Wang, Biao Yang, Bin Qian

**Affiliations:** 1Department of Automation, Faculty of Information Engineering and Automation, Kunming University of Science and Technology, Kunming 650500, China; 2Yunnan Key Laboratory of Green Energy, Electric Power Measurement Digitalization, Control and Protection, Kunming 650500, China; 3School of Chemistry and Chemical Engineering, Beijing Institute of Technology, Beijing 100081, China

**Keywords:** soft sensor, semi-supervised data streams, online clustering, adaptive switching prediction, sample augmentation, Gaussian process regression

## Abstract

In the era of big data, industrial process data are often generated rapidly in the form of streams. Thus, how to process such sequential and high-speed stream data in real time and provide critical quality variable predictions has become a critical issue for facilitating efficient process control and monitoring in the process industry. Traditionally, soft sensor models are usually built through offline batch learning, which remain unchanged during the online implementation phase. Once the process state changes, soft sensors built from historical data cannot provide accurate predictions. In practice, industrial process data streams often exhibit characteristics such as nonlinearity, time-varying behavior, and label scarcity, which pose great challenges for building high-performance soft sensor models. To address this issue, an online-dynamic-clustering-based soft sensor (ODCSS) is proposed for industrial semi-supervised data streams. The method achieves automatic generation and update of clusters and samples deletion through online dynamic clustering, thus enabling online dynamic identification of process states. Meanwhile, selective ensemble learning and just-in-time learning (JITL) are employed through an adaptive switching prediction strategy, which enables dealing with gradual and abrupt changes in process characteristics and thus alleviates model performance degradation caused by concept drift. In addition, semi-supervised learning is introduced to exploit the information of unlabeled samples and obtain high-confidence pseudo-labeled samples to expand the labeled training set. The proposed method can effectively deal with nonlinearity, time-variability, and label scarcity issues in the process data stream environment and thus enable reliable target variable predictions. The application results from two case studies show that the proposed ODCSS soft sensor approach is superior to conventional soft sensors in a semi-supervised data stream environment.

## 1. Introduction

In the process industry, real-time estimations of key quality parameters are of great importance for process monitoring, control, and optimization. However, due to technical and economic limitations, these key parameters related to product quality and process status cannot be measured online. Instead, soft sensor technology, as an important indirect measurement tool, has been widely used in the process industry. The core of soft sensors is constructing mathematical models between easy-to-measure secondary variables and a primary variable. In recent years, rapid advances in machine learning, data science, computer, and communication technologies have stimulated development of data-driven soft sensor techniques [1,2]. Typical data-driven soft sensor modeling approaches include principal component regression (PCR), partial least squares (PLS), neuro-fuzzy systems (NFS), Gaussian process regression (GPR), artificial neural networks (ANN), support vector regression (SVR), recurrent neural network (RNN), and regression generative adversarial networks model with gradient penalty (RGAN-GP) [3,4,5,6,7]. However, industrial processes often exhibit complex characteristics, such as nonlinearity, time-variability, and label scarcity, which pose great challenges for building high-performance data-driven soft sensor models.

Process data are often characterized by strong nonlinearity due to the inherent complexity of process production, variability in operating conditions, and demands for different grades of products. A popular solution to this issue is to use a local learning modeling framework. This approach is based on the idea of “divide and conquer” to describe a complex nonlinear space with locally linear spaces and build locally valid models for local regions of process. Common local soft sensor methods include clustering, ensemble learning, and JITL. Among them, clustering algorithms, such as k-means [8] and Gaussian mixture models, aim to divide process data into multiple clusters by some similarity criterion for describing different local process regions. Ensemble learning methods, such as bagging [9] and boosting [10], construct diverse weak learners and combine them to obtain a strong ensemble. The JITL method [11] is implemented through an online manner, where similar samples relevant to query samples are selected for online local models.

Another issue that needs to be addressed in soft sensor modeling is process time-variability. The data characteristics of production processes often change with time due to sensor drift, seasonal factors, and catalyst deactivation, which result in degradation of soft sensor models’ performance. In the field of machine learning, this problem is called concept drift. For such time-varying production environments, soft sensor models built offline are not well adapted to changes in process states. Therefore, it is necessary to introduce adaptive learning mechanisms to achieve self-maintenance of soft sensor models [12]. Depending on the change’s speed, time-varying features can be classified into two types: gradual and abrupt changes. Gradual changes proceed slowly, while abrupt changes rapidly shift from one state to another, which makes it difficult for the model to accommodate the changing environments and thus leads to a decrease in model prediction performance. Popularly used adaptation mechanisms include moving window, recursive update, time difference modeling, offset compensation, JITL, and ensemble learning [13,14]. Among them, the first four methods can deal with gradual changes effectively, while the last two methods are good at dealing with abrupt changes.

Moreover, scarcity of labeled samples is also a great challenge to limiting accuracy of soft sensors. Generally, soft sensor models are built through supervised learning; thus, their prediction performance relies extremely on using several labeled data. However, in actual industrial processes, it is very common to encounter the dilemma of “labeled data poor and unlabeled data rich” due to the high cost of obtaining sufficient labeled data. To tackle such a challenge, semi-supervised learning methods have been proposed, aiming to improve model performance by making full use of information from unlabeled samples [15]. The most representative semi-supervised methods are self-training, co-training, generative models, low-density region segmentation, and graph-based methods [16,17,18].

Despite availability of many soft sensor methods proposed for dealing with the problems of process nonlinearity, time-varying behavior, multi-phase/multi-mode property, and labeled sample sparsity, these approaches usually assume that abundant modeling data are available for offline modeling. In practice, this assumption has many drawbacks for practical soft sensor modeling in the process industry: (1) samples obtained offline often ignore temporal correlation; (2) there may be several samples that are not helpful for the current prediction; (3) over-focusing on historical samples while ignoring the latest samples; (4) once the prediction model is implemented, it remains unchanged and thus cannot adapt well to new state changes, which leads to model performance deterioration. In the actual process industry, it is a natural aspect that process data are generated in the form of data streams. Unlike traditional static data, data streams have characteristics of infinite, sequential, high-speed arrival, concept drift, and label scarcity [19]. Therefore, it is still a challenging issue to develop well-performing soft sensors for process data streams.

Over the years, the research on data streams has mainly focused on classification and clustering, while there are few studies on data stream regression. Among of them, the research on data stream regression mainly focuses on solving concept drift problems in nonstationary environments. The most commonly used algorithm is AMRules [20], which is the first streaming rule-learning algorithm for regression problems that learns ordered and rule-free sets from data streams. Another classical algorithm is fast incremental model tree with drift detection (FIMI-DD) [21], a method for learning a regression model tree that identifies changes in tree structure through explicit change detection and informed drift adaptation strategies. Until now, many existing data stream regression algorithms have been implemented based on the above two algorithms to improve performance while achieving better prediction results. The main work is summarized as follows.

(1)Rule-based data stream regression algorithms. Shaker et al. [22] proposed a fuzzy rule-learning algorithm called TSKstream for data stream adaptive regression. The method introduces a new TSK fuzzy rule induction strategy by combining the merits of the rule induction concept implemented in AMRules with the expressive power of TSK fuzzy rules, which solves the problem of adaptive learning from evolving data streams. Yu et al. [23] proposed an online multi-output regression algorithm called MORStreaming, which learns instances based on topological networks and correlations between outputs based on adaptive rules and can solve the problem of multiple output regression in the data stream environment.(2)Tree-model-based data stream regression algorithms. Gomes et al. [24] proposed an adaptive random forest algorithm capable of handling data stream regression tasks (ARF Reg). The algorithm uses the adaptive sliding window drift detection method and experiments with the original Page Hinkley test inside each FIMI-DD to detect and adapt to drift. Zhong et al. [25] proposed an online weight-learning random forest regression (OWL-RFR). This method focuses on a sequential dataset problem that has been ignored in most studies on online RFs and improves the predictive accuracy of the regression model by exploiting data correlation. Subsequently, Zhong et al. [26] proposed an adaptive long short-term memory online random forest regression, which designs an adaptive memory activation mechanism to handle static data streams or non-static data streams with different types of conceptual drift. Further, some researchers have attempted to introduce online clustering for dealing with data stream regression modeling. Ferdaus et al. [27] proposed a new type of fuzzy rules based on the concept of hyperplane clustering for data stream regression problems called PALM; it can automatically generate, merge, and adjust hyperplane-based fuzzy rules in a single pass, which can effectively handle the concept drift of each path in the data stream, with advantages of low memory burden and low computational complexity. Song et al. [28] proposed a data stream regression method based on fuzzy clustering called FUZZ-CARE. The algorithm can accomplish dynamic identification, training, and storage of three patterns, and the affiliation matrix obtained by fuzzy C-means clustering indicates affiliation of subsequent samples of the corresponding pattern. This method can address the concept drift problem in non-stationary environments and effectively avoid the problem of under-training due to lack of new data.

It is evident from the above studies that state identification of a process data stream is the key to obtaining high prediction accuracy from data stream regression models. For this reason, data stream clustering has been widely used to achieve local process state identification. Unlike traditional offline, single, fixed number clustering methods, data stream clustering has the advantage of online incremental learning and updating, which can provide concise representations of discovered clusters and enable processing of new samples in an incremental manner for clear and fast detection of outlier points. Generally, data stream clustering can be classified into hierarchical methods, partition-based methods, grid-based methods, density-based methods, and model-based methods [29]. Hierarchical data stream clustering algorithms use tree structure, have high complexity, and are sensitive to outliers. The representative ones are ROCK [30], evolution-based technique for stream clustering (E-stream) [31], and its extension, HUE-stream [32]. Partition-based clustering algorithms partition data into a predefined number of hyperspherical clusters, such as CluStream [33], Streamkm++ [34], and adaptive streaming k-means [35]. Grid-based clustering algorithms require determining the number of grids in advance, and they can find arbitrarily shaped clusters and are more suitable for low-dimensional data, such as WaveCluster [36], a grid-based clustering algorithm for high-dimensional data streams (GCHDS) [37], and DGClust [38]. Density-based algorithms form micro-clusters by radius and density, can find arbitrarily shaped clusters, and automatically determine number of clusters, which is suitable for high-dimensional data, and are capable of handling noise, such as DBSCAN [39], DenStream [40], online clustering algorithm for evolving data streams (CEDAS) [41], MuDi-Stream [42], and an improved data stream clustering algorithm [43]. Performance of model-based clustering algorithms is mainly influenced by the chosen model, such as CluDistream [44] and SWEM algorithm [45].

Among the above-mentioned clustering methods, density-based clustering algorithms are frequently used due to their advantages, such as not requiring the number of clusters to be determined in advance, abilities of identifying outlier points, handling noise, and finding clusters of arbitrary shapes, and their applicability to high-dimensional data. Although traditional density-based clustering algorithms for data streams can discover clusters of arbitrary shapes, the generated clusters cannot evolve and overcome unstable data streams well. To address this issue, Hyde et al. proposed an improved algorithm of CODAS [46], called CEDAS [41], which is the first fully online clustering algorithm for evolving data streams. It consists of two main phases. The first phase establishes clusters, which enable updating, generation, and disappearance of clusters, while the second phase consists of forming macro-clusters from micro-clusters, which can handle changing data streams as well as noise characteristics and provide high-quality clustering results. However, this algorithm requires radius and threshold to be defined in advance, which has a large influence on the clustering results. Thus, a method of buffer-based online clustering (BOCEDS) was proposed to automatically determine clustering parameters [47]. In addition, CEDGM has been proposed by using a grid-based approach as an outlier buffer to handle multi-density data and noise [48]. Considering the effectiveness of online clustering to overcome data stream noise and achieve high-quality clustering results, this paper aims to build on it to achieve online dynamic clustering for industrial semi-supervised data streams and thus build a high-performance data stream soft sensor model.

Despite the availability of numerous methods proposed for data stream classification and clustering problems, so far, few attempts to study soft sensor applications from the perspective of process data streams have occurred. Since it is very common that numerous unlabeled data and a small number of labeled data are generated with the process data streams in the process industry, this paper focuses on soft sensor modeling for industrial semi-supervised data streams and aims to address the following issues: (1) as with traditional soft sensor methods, data stream soft sensor models also need to effectively deal with process nonlinearity; (2) it is desirable to empower soft sensor models with online learning capabilities for capturing the latest process states to prevent model performance deterioration; (3) it is appealing to mine both historical and new data information to avoid catastrophic forgetting of historical information by the newly acquired model; (4) performance of soft sensor models needs to be enhanced by semi-supervised learning using both labeled and unlabeled data.

To solve the above-mentioned problems, an online-dynamic-clustering-based soft sensor method (ODCSS) is proposed for semi-supervised data streams. ODCSS is capable of handling nonlinearity, time-variability, and label scarcity issues in industrial data streams. Two case studies have been reported to verify the effectiveness and superiority of the proposed ODCSS algorithm. The main contributions of this paper can be summarized as follows.

(1)An online dynamic clustering method is proposed to enable online identification of process states concealed in data streams. Unlike offline clustering, this method can automatically generate and update clusters in an online manner; a spatio-temporal double-weighting strategy is used to eliminate obsolete samples in clusters, which can effectively capture the time-varying characteristics of process data streams.(2)An adaptive switching prediction strategy is proposed by combining selective ensemble learning and JITL. If the query sample is judged to be an outlier, JITL is used for prediction. Otherwise, selective ensemble learning is used. The method facilitates effective handling of both gradual and abrupt changes in process characteristics, which enables preventing high soft sensor performance from deteriorating in time-varying environments.(3)Online semi-supervised learning is introduced to mine both labeled and unlabeled sample information, thus expanding the labeled training set. This strategy can effectively alleviate the problem of insufficient labeled modeling samples and can obtain better prediction performance than supervised learning.

The rest of the paper is organized as follows. Section 2 provides details of the proposed ODCSS approach. Section 3 demonstrates the effectiveness of the proposed method through two case studies. Finally, Section 4 concludes the paper. A brief introduction of GPR, self-training, and JITL can be found in Appendix A, Appendix B, Appendix C, respectively.

## 2. Proposed ODCSS Soft Sensor Method for Industrial Semi-Supervised Data Streams

Soft sensor modeling for data stream remains challenging for the following reasons. First, process data are often characterized by strong nonlinearity and time-variability, which makes the linear and nonadaptive models function badly. Second, many current data stream regression approaches rely on a single-model structure, thus limiting their prediction accuracy and reliability. Third, in industrial processes, it is often the case that labeled data are scarce but unlabeled data are abundant. In such situations, conventional supervised data stream regression models are ill-suited for semi-supervised data streams. Therefore, we propose a new soft sensor method, ODCSS, for industrial semi-supervised data streams. The main steps of ODCSS include: (1) online dynamic clustering; (2) adaptive switching prediction; and (3) sample augmentation and maintenance. The details are described in the following subsections.

### 2.1. Problem Definition

Semi-supervised data streams. Assuming that the data streams are the continuous sequence containing n (n→∞) instances, i.e., D = {$s_{0}$, $s_{T}$, $s_{2T}$, …, $s_{t-T}$, $s_{t}$}, where $s_{t}$ is the sample arriving at time t and $s_{t}$ = {$x_{t}$,$y_{t}$}, where $x_{t}$ denotes input features, $y_{t}$ is the label of the sample. In the context of soft sensor applications, $y_{t}$ corresponds to the hard-to-measure variables in industrial processes, such as product concentration, catalyst activity, etc. In the data streams, the ideal situation is that all data are labeled, which allows us to perform supervised learning. However, it is often expensive and laborious to obtain labels, thus creating a mixture of a small number of labeled samples and a large number of unlabeled samples, which are called semi-supervised data. Therefore, we assume that the data streams consist of successive arrivals of high-frequency unlabeled samples and low-frequency labeled samples, as denoted by $D_{t}$ = {$s_{0}^{l}$, $s_{T}^{u}$, …, $s_{mT-T}^{u}$, $s_{mT}^{l}$, $s_{mT+T}^{u}$, …, $s_{t-T}^{u}$, $s_{t}^{l}$}, where $s_{*}^{u}$ and $s_{*}^{l}$ represent unlabeled and labeled samples, respectively.

Regression task for soft sensor modeling. Suppose semi-supervised data streams $D_{t}$ has been obtained up to time t. Given an online obtained $x_{t}$, unknown label $y_{t}^{u}$ is required to be estimated. Thus, mathematical model $f_{t}$ should be constructed based on coming semi-supervised data streams $D_{t}$; that is,
(1)$${\hat{y}}_{t}^{u}=f_{t}\left(x_{t}\right)$$

As can be seen from Equation (1), soft sensor modeling for a data stream has the following characteristics.

(1)Modeling data $D_{t}$ changes and accumulates over time. After a period of time, numerous historical data and a small set of recent process data will be obtained. Thus, it is crucial to coordinate the roles of historical data and the latest data for soft sensor modeling. If only the latest information is addressed while ignoring historical valuable information, the generalization performance of the model cannot be guaranteed. Contrarily, focusing only on historical information will make the soft sensor model unable to capture the latest process state.(2)In most cases, the true value of ${\hat{y}}_{t}^{u}$ is unknown, and only a few observed values are obtained through low-frequency and large-delay analysis. Traditional supervised learning can only effectively use labeled samples, ignoring the information from unlabeled samples. In practice, unlabeled samples also contain rich information about the process states. Thus, it is also an important way to improve soft sensor models by fully exploiting labeled and unlabeled samples through a semi-supervised learning framework.(3)Stream data $D_{t}$ usually implies a complex nonlinear relationship between inputs and outputs. Therefore, the idea of local learning is often considered to obtain better prediction performance than a single global model. In addition, as the process runs and $D_{t}$ changes, $f_{t}$ is not constant and often exhibits significant time-varying characteristics. Therefore, to prevent degradation of the prediction performance of $f_{t}$, it is necessary to introduce a suitable adaptive learning mechanism to achieve an online update of $f_{t}$.

### 2.2. Online Dynamic Clustering

Industrial process stream data $D_{t}$ often contains rich process state information; however, traditional global modeling is difficult to obtain high prediction accuracy because local process states cannot be well characterized. For this reason, clustering algorithms are often used to implement process state identification. However, traditional clustering algorithms are usually implemented offline and the resulting clusters remain unchanged once the clustering is completed. Such approaches are not suitable for handling data streams that evolve in real time. Thus, in data stream environments, local process state identifications need to be performed dynamically in an online manner. To this end, an online dynamic clustering (ODC) method based on density estimation is proposed to achieve online state identification of process data streams.

Traditionally, offline clustering algorithms for batch data can usually obtain multiple clusters at the same time. In contrast to this, ODC processes the data online one by one and assigns them to the appropriate clusters. Without loss of generality, Euclidean distance is chosen to measure the similarity between samples in this paper. The calculation formula is as follows:(2)$$d\left(x_{i},x_{j}\right)=\|x_{i}-x_{j}\|$$
where $x_{i}$ and $x_{j}$ represent two arbitrary samples.

#### 2.2.1. Initialization

The ODC process requires setting two important initial parameters: cluster radius R and minimum density threshold M. Given a dataset, the most appropriate cluster radius needs to be selected based on the data features, i.e., the maximum allowable distance from the cluster center to the cluster edge. When the distance between data points is less than the radius and the number of data reaches the minimum density threshold, cluster C can be formed. The average of all sample features in the cluster is calculated as cluster center c. Clustering is an unsupervised process and is completed using features only, where clustering center c is simply calculated as
(3)$$c=\left(\overset{n}{\underset{i=1}{\sum }}x_{i}\right)/n$$
where n is the sample size in the cluster and $x_{i}$ is the i-th sample.

The first cluster is constructed to store information related to the clusters, including clustering center c and the data samples stored in the cluster. Following this, online dynamic clustering is performed for sequentially arrived query sample $x_{t}$.

#### 2.2.2. Updating the Cluster

Assume that multiple clusters have been obtained, i.e., ${\left\{C_{t}^{m}\right\}}_{m=1}^{M}$, and a new query sample $x_{t}$ arrives; then, Euclidean distances d between $x_{t}$ and the existing cluster centers are calculated as
(4)$$d_{t}={\left\{d_{t}^{m}\right\}}_{m=1}^{M}$$

If $d_{t}^{m}\leq R$, $x_{t}$ is included to cluster $C_{t}^{m}$. Considering the boundary fuzzy property between different process states, a softening score strategy is used to group the sample points into all eligible clusters. In addition, if $d_{t}^{m}\leq 2R/3$, cluster center c will be updated to accommodate concept drift:(5)$$\left\{\begin{array}{c} x_{t}\in C_{t}^{m},d_{t}^{m}\leq R\\ c^{\prime }=\left(c*\left(n-1\right)+x_{t}\right)/n,d_{t}^{m}\leq 2R/3 \end{array}\right.$$
where $C_{t}^{m}$ is the m-th cluster, $d_{t}^{m}$ is the Euclidean distance between the new sample and the m-th cluster, n is the number of samples in the m-th cluster, and c and $c^{\prime }$ denote the cluster centers before and after updating.

It should be noted that the shape of the cluster is not fixed but further evolves with migration of cluster center c.

#### 2.2.3. Generating the New Cluster

Since the process state is always changing, the clusters need to adapt to new state changes as samples accumulate. Thus, it is desirable to generate new clusters to accommodate concept drift as the process data grow. If Euclidean distance d between $x_{t}$ and cluster center c is larger than the radius, this sample is regarded as an outlier. In such a case, distances $d_{t}^{o}$ between the existing outliers are calculated, and a new cluster is generated if the number of outliers with distances less than the radius reaches minimum density threshold M. The center of the new cluster is calculated using Equation (3). The remaining outliers that do not form clusters are retained and exist separately in space.

#### 2.2.4. Removing Outdated Data in the Cluster

Ideally, data streams are an infinitely increasing process. However, the update burden of clusters as well as the computational efficiency of soft sensor models will grow as the process data accumulates. Hence, it is appealing to remove the outdated samples from the clusters.

Since Euclidean distance only considers spatial similarity and tends to ignore the temporal relevance of samples, a spatio-temporal double weighting strategy is proposed to consider both spatial and temporal similarity between historical and recent samples. For this purpose, the spatio-temporal weights are calculated to eliminate the least influential samples in the cluster on the query sample $x_{t}$:(6)$$w_{i}={\left(\alpha *\left(d_{i}/\sum_{i=1}^{n}d_{i}\right)+\left(1-\alpha \right)*\left(t_{i}/\sum_{i=1}^{n}t_{i}\right)\right)}^{-1},\;i=1,2,\ldots ,n$$
where $d_{i}$ is the distance from the i-th sample in the cluster to the cluster center and $t_{i}$ is the time interval between the i-th sample in the cluster and query sample $x_{t}$, and α is a parameter controlling the influence of spatio-temporal information. The smaller the weights are, the smaller the corresponding history samples have an influence on query sample $x_{t}$. The spatio-temporal weights are sorted in ascending order, and a fixed proportion of historical samples are removed.

The pseudo code for the ODC method is given in Algorithm 1.
**Algorithm 1:** Online dynamic clustering (ODC) **INPUT**: $D_{t}$: Data streams      R: Cluster radius      M: Minimum density threshold       α: Controlling parameter       P: The percentage of removed samples**PROCESS:**1: Create a cluster structure containing:   **%% Initialization**2:    C(c) = $x_{1}$;  %% Cluster center3:    C(Data) = $x_{1}$;  %% Save data4:    C(O) = [];  %% Save outlier, initially empty5: Calculate distances $d_{t}$ between $x_{t}$ and all cluster centers c using Equation (3).6: **if** $d_{t}\leq R$   **%% Updating the cluster**7:    $x_{t}$ is stored in all clusters where $d_{t}\leq R$;8:    **if**
$d_{t}\leq$2R/3 9:     Update cluster center c using Equation (5);10:     **end if**11: **else if** $d_{t}>R$   **%% Generating a new cluster**12:     $x_{t}$ is stored in the outliers;13:     Calculate distances $d_{t}^{o}$ between samples stored in outliers;14:     **if** $d_{t}^{o}\leq R$ & size(outliers) ≥M
15:     Generate a new cluster;16:     $x_{t}$ is stored in the new cluster;17:     Calculate new cluster center c using Equation (3);18:     **end if**19: **end if****%% Removing the outdated data in the cluster**20: Calculate the spatio-temporal weights for each sample in the cluster using Equation (6);21: Sort w in ascending order;22: According to the sorting, delete the samples with the smallest weights in the cluster and fix the proportion of the deleted samples to P;**OUTPUT**: Cluster results C

### 2.3. Adaptive Switching Prediction

By applying online dynamic clustering, query sample $x_{t}$ can be assigned to either existing clusters or outliers. In comparison with samples within clusters, outliers reveal significantly different statistical characteristics of process variables. Thus, an adaptive switching prediction method is proposed by combining adaptive selective ensemble learning with JITL to achieve real-time predictions for within-cluster samples and outliers, respectively. In addition, GPR is used as the base modeling technique, which is a nonparametric regression model that can learn arbitrary forms of functions with advantages such as smoothness, parameter adaption, and strong capability of fitting nonlinear data.

#### 2.3.1. Adaptive Selective Ensemble Learning for Online Prediction

Suppose that $M_{t}$ clusters have been obtained at moment t, for which $M_{t}$ GPR base models ${\left\{f_{t}^{m}\right\}}_{m=1}^{M}$ are built. When query sample $x_{t}$ arrives, prediction is achieved using an adaptive selective ensemble learning strategy if $x_{t}$ is classified into clusters. Three main key steps are described as follows:

Step 1: evaluate distances d between $x_{t}$ and its corresponding cluster center c, and select $m_{t}$($m_{t}\leq M_{t}$) GPR models ${\left\{f_{t}^{sel,m}\right\}}_{m=1}^{m_{t}}$ with small distances d.

Step 2: provide $m_{t}$ prediction values {${\hat{y}}_{t,1}$, ${\hat{y}}_{t,2}$, …, ${\hat{y}}_{t,m_{t}}$} based on the obtained models, and use simple averaging rule to obtain final prediction output ${\hat{y}}_{t}$:(7)$${\hat{y}}_{t}=(\overset{m_{t}}{\underset{i=1}{\sum }}{\hat{y}}_{t,i})/m_{t}$$

Step 3: if a new labeled sample or high-confidence pseudo-labeled sample is added to the clusters, the corresponding GPR models will be rebuilt.

It is worth noting that new labeled samples are often obtained by offline analysis, while pseudo-labeled samples are obtained by self-training, which is detailed in Section 2.4.

Figure 1 shows the schematic diagram of the adaptive selective ensemble learning framework.

#### 2.3.2. Just-In-Time Learning for Online Prediction

If query sample $x_{t}$ is judged to be an outlier, JITL is used for prediction. Since the outliers are samples deviating from the clusters, if the outliers are predicted by using the models built from clusters, the predictions may deviate greatly from the actual values. Therefore, by using all labeled samples as the database, a small-size dataset $D_{\mathrm{simi}}=\left\{X_{\mathrm{simi}},y_{\mathrm{simi}}\right\}$ similar to query sample $x_{t}$ is constructed to build a JITGPR model for online prediction of ${\hat{y}}_{t}$.

Thus far, various similarity measures have been proposed for JITL methods [49], including Euclidean distance similarity, cosine similarity, covariance weighted similarity, Manhattan distance similarity, Pearson coefficient similarity, etc.

Algorithm 2 presents the pseudo code of the adaptive switching prediction process.
**Algorithm 2:** Adaptive switching prediction**INPUT**: $D_{t}$: Data streams      C: Online clustering results      m: The maximal size of the selective clusters **PROCESS**: 1: **if** $x_{t}$ is within clusters    **%% Adaptive selective ensemble learning for online prediction** 2:     Select the GPR models ${\left\{f_{t}^{sel,m}\right\}}_{m=1}^{m_{t}}$ corresponding to the closest $m^{t}$ clusters; 3:     Predict $x_{t}$ using the selected models ${\left\{f_{t}^{sel,m}\right\}}_{m=1}^{m_{t}}$ to obtain {${\hat{y}}_{t,1}$, ${\hat{y}}_{t,2}$, …, ${\hat{y}}_{t,m_{t}}$}; 4:     Calculate the average of ${\hat{y}}_{t,i}$ using Equation (7) to obtain final prediction output ${\hat{y}}_{t}$; 5:     **for** i = 1, 2, …, m
**do** 6:      **if** there is an update to the samples in the i-th cluster 7:       Rebuild a new GPR model using the updated samples; 8:      **else if** the samples in the i-th cluster are not updated 9:       Keep the old GPR model; 10:    **end if** 11:   **end for** 12: **else if** $x_{t}$ is judged to be an outlier    **%% Just-in-time learning for online prediction** 13:   Select the most similar samples to the $x_{t}$ as training set $D_{\mathrm{simi}}$ from the historical labeled samples; 14:   Build a JITGPR model with $D_{\mathrm{simi}}$; 15:   Predict $x_{t}$ using the JITGPR model; 16:   Obtain finally predicted result ${\hat{y}}_{t}$; 17: **end if****OUTPUT**: Prediction result ${\hat{y}}_{t}$

### 2.4. Sample Augmentation and Maintenance

Although the production process produces a large number of data records in the form of streams, the proportion of labeled samples is small. In practice, for arbitrary query sample $x_{t}$, its label can be estimated by using the adaptive switching prediction method. Such predictions are called pseudo labels, which can be used to update the model if they are highly accurate. However, the actual labels for most unlabeled samples are unknown due to absence of offline analysis. For this reason, we borrow the idea of self-training, a widely used semi-supervised learning paradigm, to obtain high-confidence pseudo-labeled samples and then update the models.

One main difficulty of self-training is defining confidence evaluation criteria for selecting high-quality pseudo-labeled samples. Thus, we attempt to evaluate improvement of prediction performance before and after introducing pseudo-labeled samples. The specific steps are described as follows:

Step 1: select a certain proportion of the labeled samples similar to query sample $x_{t}$ as the online validation set and use the remaining labeled samples as the online training set.

Step 2: build two GPR models based on the training set before and after adding the pseudo-labeled data {$x_{t}$, ${\hat{y}}_{t}^{u}$}, respectively.

Step 3: evaluate the prediction RMSE values of the two models on the validation set, and then the improvement rate (IR) can be calculated as:(8)$$IR=\frac{\mathrm{RMSE}-{\mathrm{RMSE}}^{\prime }}{\mathrm{RMSE}}$$
where RMSE and ${\mathrm{RMSE}}^{\prime }$ are the root mean square errors of the GPR model on the validation set before and after the pseudo-labeled sample is added to the training set.

Step 4: if the IR value of the pseudo-label ${\hat{y}}_{t}^{u}$ is greater than confidence threshold $IR_{th}$, {$x_{t}$, ${\hat{y}}_{t}^{u}$} is added to the corresponding cluster to update the training set. Otherwise, this sample is removed from the clusters.

Although the latest pseudo-labeled samples added to the clusters can improve the prediction performance for the query sample, accumulating too many pseudo-labeled samples can cause error accumulation, so timely deletion of the outdated historical samples is very essential for reducing prediction deviations. To make full use of the information from the recent unlabeled samples, the sample deletion procedure is started only when the latest true label is detected, and the pseudo-labeled data that have the least impact on query sample $x_{t}$ are deleted. The above process is accomplished by online dynamic clustering, which can reduce the update burden of clustering on the one hand and improve the prediction efficiency of soft sensor models on the other hand.

### 2.5. Implementation Procedure of ODCSS

The overall framework of the ODCSS soft sensor method is illustrated in Figure 2. With the process data arriving in the form of stream, ODCSS is implemented mainly through three steps: online dynamic clustering, adaptive switching prediction, and sample augmentation and maintenance.

Given the input: (1) data streams $D_{t}$={$s_{0}^{l}$, $s_{T}^{u}$, …, $s_{mT-T}^{u}$, $s_{mT}^{l}$, $s_{mT+T}^{u}$, …, $s_{t-T}^{u}$, $s_{t}^{l}$}; and (2) modeling parameters: clustering radius R, minimum density threshold M, controlling parameter α, the proportion of deleted data P, confidence threshold $IR_{th}$, and the maximal ensemble size m. Assume that $M_{t}$ clusters ${\left\{C_{t}^{m}\right\}}_{m=1}^{M}$ and their corresponding models ${\left\{f_{t}^{m}\right\}}_{m=1}^{M}$ have been constructed at moment t, along with a small number of outliers. The following steps are repeated for any newly arrived query sample:

Step 1: when query sample $x_{t}$ arrives at time t, online dynamic clustering is performed to include $x_{t}$ to clusters or recognize $x_{t}$ as an outlier.

Step 2: if $x_{t}$ belongs to clusters, first select the GPR models corresponding to the nearest $m_{t}$($m_{t}\leq M_{t}$) clusters, and then obtain a set of predicted values {${\hat{y}}_{t,1}$, ${\hat{y}}_{t,2}$, …, ${\hat{y}}_{t,m_{t}}$} based on the selected GPR models; finally, calculate the average value of predicted values to obtain final prediction output ${\hat{y}}_{t}$. If $x_{t}$ belongs to outliers, use a JITGPR model to obtain final prediction output ${\hat{y}}_{t}$.

Step 3: the confidence of {$x_{t}$,${\hat{y}}_{t}^{u}$} in the cluster is evaluated based on the proposed strategy in Section 2.4. If IR exceeds $IR_{th}$, the obtained pseudo-labeled sample is added to the clusters to update the models. Otherwise, this sample is discarded.

Step 4: when actual label $y_{t}$ of $x_{t}$ is available, the sample {$x_{t}$,$y_{t}$} is used to update the training set and base models, while the corresponding pseudo-labeled sample is removed. Meanwhile, the outdated samples are removed by using the proposed ODC method.

## 3. Case Studies

### 3.1. Methods for Comparison

In this section, the proposed ODCSS soft sensor method is evaluated through applications to the Tennessee Eastman (TE) chemical process and an industrial fed-batch chlortetracycline (CTC) fermentation process. The compared methods are as follows:(i)MWGPR: moving window Gaussian process regression model.(ii)JITGPR [49]: just-in-time learning Gaussian process regression.(iii)OSELM [50]: a sequential learning algorithm called online sequential extreme learning machine, which can learn not only the training data one by one but also block by block (with fixed or varying length).(iv)OSVR [51]: online support vector machines for regression, which achieves incremental updating through moving window strategy.(v)PALM [27]: parsimonious learning machine, a data stream regression method, which utilizes new fuzzy rules based on the concept of hyperplane clustering. It can automatically generate, merge, and adjust the fuzzy rules based on the hyperplane. The authors propose two types of PALM models, type-1 and type-2, each of which can be divided into local and global updating strategies. To get closer to the idea of local modeling in this paper, we select type-2 PALM with better performance and local update strategy as the comparison method.(vi)OSEGPR: online selective ensemble Gaussian process regression. The basic idea of this approach is that, assuming m GPR models have been established by time t, when a new query sample comes, a global GPR model is established using all already obtained historical samples, then the prediction performance of all retained models on an online validation set. Next, part models with high performance are selected to provide the ensemble prediction results. The above process is repeated as new query samples arrive.(vii)SS-OSEGPR: semi-supervised online selective ensemble Gaussian process regression, which introduces unlabeled samples to OSEGPR. Using the confidence evaluation strategy in Section 2.4 of this paper, we select pseudo-labels with high confidence to expand the training set and update the model.(viii)ODCSS_S_: a degenerated version of the proposed online-dynamic-clustering-based soft sensor modeling for industrial supervised data streams. That is, the online soft sensor modeling process is completed using only the labeled data streams.(ix)ODCSS: the proposed online-dynamic-clustering-based soft sensor modeling for industrial semi-supervised data streams.

### 3.2. Experimental Setup and Evaluation Metrics

In order to obtain high-performance prediction results for each soft sensor model, the key model parameters need to be chosen carefully. Especially, the number of modeling samples for the compared methods are set to the same as the number of initial training samples for ODCSS, including the width of moving window for MWGPR, the size of local modeling samples for JITGPR, the size of initial training samples for OSELM, and the size of online validation data for OSEGPR and SS-OSEGPR. In addition, with reference to [50], the prediction block in OSELM is set to 1, and the number of hidden neurons should be smaller than the initial number of training samples. The parameter setting of PALM method refers to [27]. The two parameters of *R* and *M* related to clustering in the ODCSS method are determined according to [41], which are adjusted according to different application scenarios. The remaining model parameters are determined within a reasonable range through the trial-and-error method. Moreover, for JITGPR, OSEGPR, and SS-OSEGPR, the best similarity measure is selected from Euclidean distance similarity, cosine similarity, covariance weighted similarity, Manhattan distance similarity, and Pearson correlation coefficient similarity, whose definitions can be found in [49]. Further, the Matern covariance function with noise term is used for all GPR based models.

To evaluate the prediction performance of soft sensor models, the following evaluation metrics are considered, including root mean square error (RMSE), mean absolute error (MAE), mean absolute percentage error (MAPE), and coefficient of determination $(R^{2}$) [52]. Among them, RMSE and MAE are used to measure the closeness between the predicted and true outputs. MAPE is a measure of the variation of dependent series from its model-predicted level. The smaller the MAPE, the better the model performance, and, for perfect fit, the value of MAPE is zero [53]. $R^{2}$ is the square of Pearson’s correlation coefficient. It represents a squared correlation between the actual output and the predicted output and measures how much of the total variance in the output variable data can be explained by the model. The closer $R^{2}$ is to 1, the better the performance of the models. Usually, if the value of $R^{2}$ is greater than 0.5, the model predictions can be judged as satisfactory [54]. The above evaluation metrics are defined as follows:(9)$$\mathrm{RMSE}=\sqrt{\frac{1}{n_{test}}\overset{n_{test}}{\underset{i=1}{\sum }}{\left({\hat{y}}_{test,i}-y_{test,i}\right)}^{2}}$$
(10)$$\mathrm{MAE}=\frac{1}{n_{test}}\overset{n_{test}}{\underset{i=1}{\sum }}\left|{\hat{y}}_{test,i}-y_{test,i}\right|$$
(11)$$\mathrm{MAPE}=\frac{1}{n_{test}}\overset{n_{test}}{\underset{i=1}{\sum }}\left|\frac{{\hat{y}}_{test,i}-y_{test,i}}{y_{test,i}}\right|\times 100\%$$
(12)$$R^{2}=1-\frac{\sum_{i=1}^{n_{test}}{\left({\hat{y}}_{test,i}-y_{test,i}\right)}^{2}}{\sum_{i=1}^{n_{test}}{\left(y_{test,i}-{\bar{y}}_{test}\right)}^{2}}$$
where $n_{test}$ represents the number of test samples, $y_{test,i}$ and ${\hat{y}}_{test,i}$ are the actual and predicted values of the ith test sample, respectively, and ${\bar{y}}_{test}$ is the mean value of the test outputs.

The computer configurations for experiments are as follows. OS: Windows10 (64 bit); CPU: Inter (R) Core (TM) @ i7-10700 (2.90 GHz × 2); RAM: 16.00 GB; and MATLAB version: 2018b.

### 3.3. Tennessee Eastman (TE) Process

#### 3.3.1. Process Description

TE chemical process has been widely used to test control, monitoring, and fault diagnosis models [55]. The TE process flow diagram is shown in Figure 3, which mainly consists of several operating units, such as a continuous stirred reactor, splitting condenser, gas–liquid separation tower, vapor extraction tower, reboiler, and centrifugal compressor. Three gas reactors directly enter the reactor through A, D, E, and C and a certain amount of feed A enters through the condenser. The numbers 1–13 are the stream orders, representing A feed, B feed, C feed, D feed, Stripper, reactor feed, reactor product, circulation, purification, separation liquid, product, condenser water and condenser water, respectively.

TE process involves a total of 12 manipulated variables and 41 process variables, and 22 of the process variables are easy to measure and the remaining 19 are difficult to measure. It should be noted that the sampling interval for 22 process variables is 3 min, whereas, for 19 difficult variables, it is 6 min. To validate the performance of the proposed soft sensor model, the input variables selected for this case study are listed in Table 1, which includes 23 easily measured process variables and 9 manipulated variables, taking the E component in stream 11 as the primary variable under the conditions with G/H mass ratio being 40/60 and the production rate being 19.45 $m^{3}/h$. The obtained data are further divided chronologically into two subsets: the initial training set with 50 labeled samples and 1601 samples arriving online to simulate process data streams, including 1200 unlabeled samples and 451 labeled samples. Note that both the labeled and unlabeled samples from the data streams are used for online modeling and prediction, whereas only the labeled samples are used to assess prediction performance of soft sensor models.

#### 3.3.2. Parameter Settings

The optimal parameters for different algorithms are set as follows:(i)MWGPR: the width of the moving window is set to 50.(ii)JITGPR: the number of local modeling samples is set to 50, and the best similarity is covariance weighted similarity.(iii)OSELM: prediction block is set to 1 to provide one prediction value at a time, the number of hidden neurons is set to 45, and the number of initial training samples used in the initial phase is set to 50.(iv)OSVR: penalty parameter C is set to 10, tuning parameter for kernel function g is set to 0.01, and precision threshold p is set to 0.001.(v)PALM: the rule merging mechanism involves parameters $b_{1},b_{2},c_{1}$, and $c_{2}$. $b_{1}$ and $b_{2}$ are used to calculate the angle and distance between two interval-valued hyperplanes, which are set to 0.02 and 0.01, respectively. $c_{1}$ and $c_{2}$ are thresholds for the rule merging conditions defined in advance, which are set to 0.01. The remaining parameters are set as in the original paper.(vi)OSEGPR: the number of online validation samples is set to 50, the ensemble size is set to 5, and Manhattan distance similarity is chosen.(vii)SS-OSEGPR: the number of online validation samples is set to 50, the ensemble size is set to 4, the confidence threshold for selecting pseudo-labels is set to 0.03, and Euclidean distance similarity is chosen.(viii)ODCSS_S_: clustering radius R is set to 8, minimum density threshold M is set to 10, and maximal ensemble size m is set to 2.(ix)ODCSS: clustering radius R is set to 9, minimum density threshold M is set to 10, controlling parameter α is set to 0.4, proportion of deleted data P is set to 0.5, confidence threshold $IR_{th}$ is set to 0.1, and maximal ensemble size m is set to 2.

#### 3.3.3. Prediction Results and Discussion

Table 2 compares the best prediction performance of different soft sensor methods. OSELM has the highest RMSE, MAE, MAPE, and the lowest $R^{2}$, implying the worst performance. This is mainly because the method has poor local learning ability and cannot effectively characterize the local characteristics of the process. In contrast, JITGPR, MWGPR, and OSVR adopt the idea of local modeling and have stronger capability of handing local process features, so their prediction accuracy is significantly improved. Although PALM also has the ability of online dynamic clustering, the method cannot well handle abrupt-change concept drift and thus provides poor performance. Unlike the single-model methods, OSEGPR predicts by combining various global models with different performance, but its performance is only comparable to that of MWGPR and OSVR, which is mainly due to insufficient local process characterization. Compared with OSEGPR, SS-OSEGPR introduces semi-supervised learning and selects high-confidence pseudo-labels to expand the labeled training set, thus improving model performance to some extent, but its prediction errors are still high. Among all the compared methods, the proposed ODCSS method provides the lowest RMSE, MAE, MAPE, and the highest $R^{2}$. In comparison, ODCSS obtains better results than ODCSS_S_, mainly due to introduction of semi-supervised learning. Overall, when using RMSE as the baseline, the prediction accuracy of the proposed ODCSS method compared to MWGPR, JITGPR, OSELM, OSVR, PALM, OSEGPR, and SS-OSEGPR is enhanced by 20%, 25.7%, 48.7%, 18.7%, 29.8%, 17.8%, and 15.2%, respectively. After adding pseudo-labeled samples, ODCSS shows a performance improvement of 3.6% compared to ODCSS_S_. The excellent performance of ODCSS is mainly attributed to four aspects. First, online dynamic clustering helps to effectively achieve local representation of complex process features. Second, adaptive switching prediction can effectively deal with gradual- and abrupt-change concept drift and can effectively overcome the problem of model degradation. Third, the adaptive selective ensemble strategy can maximize use of information of historical samples and the latest samples, while JITL is good at addressing predictions corresponding to outliers. Fourth, introduction of semi-supervised learning can make full use of information of unlabeled samples and thus improve the performance of the model.

Figure 4 shows the prediction trends of different soft sensor methods for E component. As can be seen in Figure 4c, OSELM has the largest prediction deviations from the actual values throughout the prediction zone, especially in the later period. In contrast, MWGPR (Figure 4a), JITGPR (Figure 4b), OSVR (Figure 4d), PALM (Figure 4e), OSEGPR (Figure 4f), and SS-OSEGPE (Figure 4g) have smaller deviations, but significant drifts still exist. The proposed ODCSS_S_ (Figure 4h) and ODCSS (Figure 4i) obtain smoother predictions, further reducing the deviations. Moreover, ODCSS obtains better agreement between the predicted and actual values compared to ODCSSS. These results can also confirm the superiority of the proposed method.

In the ODCSS method, online dynamic clustering is the key to realizing online local learning, which is crucial to ensure model performance. To graphically illustrate the clustering process, the 32 input variables of the TE data are dimensionally reduced using PCA to represent the dynamic clustering process of TE by three-dimensional variables, as shown in Figure 5. The red triangles in the figure indicate the outliers, and the blue and pink circles represent the two clusters formed. Figure 5a shows the first cluster and 3 outliers formed when the 15th sample arrives. When the 29th sample point arrives, some outliers are accumulated, as shown in the red triangle in Figure 5b. The 30th sample point arrives and the outliers within the radius accumulate to the set threshold, and a new cluster is formed from the outliers, as shown by the pink point in Figure 5c. As time increases, new samples are accumulated in the already built clusters, while some of the less influential historical unlabeled samples are removed, as shown in Figure 5d–f. Among them, Figure 5f shows the final clustering graph formed after the last sample arrives. As can be seen from the figure, the final number of samples in the cluster is not very large because almost all the labeled samples are retained, while several pseudo-labeled samples are gradually eliminated in the clustering process through spatio-temporal weighting.

To ensure the efficiency of the online dynamic clustering algorithm, the clustering radius and threshold should be determined carefully. For this purpose, as shown in Table 3, we evaluated the prediction performance of ODCSS with the combination of clustering radius $R\in \left[8,9,10,11,12,13\right]$ and minimum density threshold $M\in \left[10,12,14\right]$ after fixing the other four parameters, i.e., controlling parameter α to 0.05, the ratio of the amount of deleted data P to 0.4., the confidence threshold $IR_{th}$ to 0.1, and the maximum ensemble size m to 2. The overall performance of the proposed method is better than the comparison methods; that is, within a reasonable range of parameters, this method can overcome the influence of parameter changes on the prediction performance and has better stability.

As a data-streams-oriented soft sensor, ODCSS performs model building and maintenance and target variable prediction in an online manner. As new samples are accumulated, its prediction performance gradually changes. Figure 6 presents the evolving trends of the cumulative predicted RMSE using different soft sensor methods, that is, the predicted RMSE of the test samples from the first to the current prediction. Not surprisingly, in the early stage of prediction, large prediction errors are observed for all methods due to insufficient labeled samples. In particular, OSELM has poor performance throughout the prediction process. With accumulation of labeled samples, the prediction performance of the remaining methods is improved. It is worth noting that JITGPR and MWGPR obtain good prediction performance in the range of about 50–260 of the test samples but show large error growth of 260–400. The prediction performance of PALM starts degenerating at about the 50th test sample. In contrast, both ODCSS_S_ and ODCSS have the smallest prediction errors throughout the prediction process. Comparing ODCSS_S_ with ODCSS, we can observe that ODCSS_S_ shows an increase in prediction errors during the stage of 370–400 samples, while ODCSS maintains a low prediction error all the time. These results fully illustrate the prediction accuracy and reliability of ODCSS.

To further assess whether there are significant differences between ODCSS and other methods, the Statistical Tests for Algorithms Comparison (STAC) platform [56] is applied. For this purpose, the test set is equally divided into 15 groups in chronological order to obtain the prediction RMSE values on each group. Based on these RMSE values, a non-parametric Freidman test with a Finner post hoc method is performed. Especially, with ODCSS as the control method and the significance level as 0.05, Friedman test is conducted on the group RMSE values of different methods and the statistical test results are provided in Table 4. According to the principle of Freidman test, null hypothesis $H_{0}$ in this experiment is that there is no difference between the compared methods. The *p*-value indicates the probability of supporting hypothesis $H_{0}$. When the *p*-value is less than 0.05, the statistical results show that null hypothesis $H_{0}$ is rejected, which means that there is a difference between the compared methods. As shown in Table 4, MWGPR, JITGPR, OSELM, OSVR, PALM, OSEGPR, and SS-OSEGPR reject $H_{0}$ hypothesis, which indicates that the prediction performance of ODCSS is remarkably different from these methods in TE process. In addition, we can also observe that ODCSS_S_ accepts $H_{0}\;$ hypothesis. This is mainly because it is a degraded version of the proposed ODCSS method without adding pseudo-labeled data, and introduction of semi-supervised learning has not gained significant performance improvement.

Additionally, to further explore the performance of the compared methods at different stages of testing, Table 5 lists the RMSE performance of the compared methods on a subset of testing set and ranks the performance of the proposed ODCSS method. It can be seen that ODCSS has poor prediction performance in the early stage (testing subsets 1–6), and gradually (testing subsets 11–15) achieves the best prediction results. The main reason is that, in the early stage of prediction, the proposed method contains few data in the clusters formed by local process state identification, and some samples are independent in space in the form of outliers, so the local model established does not have enough learning ability, thus resulting in poor prediction performance. With accumulation of data volume, the proposed ODCSS method gradually learns different local process states, so the best prediction performance is achieved in the late stage of prediction. These results imply that the proposed method has strong learning capability in data stream environments.

### 3.4. Industrial Fed-Batch Chlortetracycline (CTC) Fermentation Process

#### 3.4.1. Process Description

With development of science and technology in pharmaceutical, food, biological, and chemical industries, as well as agriculture, microbial fermentation has made a great impact on human daily life. As one feed antibiotic additive, chlortetracycline has become the most used bacterial growth promoter in the farming industry due to its advantages of bacterial inhibition, growth promotion, high feed utilization, and low drug residues. At the same time, with expansion of production demand and scale, enterprises have implemented higher requirements for automation and the intelligence level of the fermentation process.

Figure 7 shows the flow chart of the CTC production process. CTC fermentation is an intermittent production process, and each batch of fermentation takes 80–120 h, which mainly occurs through batch and fed-batch operation stages. Many parameters for this process have been measured online using hardware sensors; however, the biomass concentration, substrate concentration, amino nitrogen concentration, and viscosity are usually not available online and can only be analyzed through offline sampling.

In this paper, the CTC fermentation process data from Charoen Pokphand Group are used for model evaluation, where substrate concentration is used as the difficult-to-measure variable and variables listed in Table 6 are used for auxiliary variables. With an online sampling interval of 5 min and offline analysis interval of 4 h, a total of 15 batches of data from the same fermenter were collected for experiments. The first two batches of labeled samples, including a total of 43 samples, are taken as the initial training set, while the remaining 13 batches of data are used for online prediction, including 1015 unlabeled samples and 324 labeled samples. The labeled and unlabeled samples are used for online modeling and prediction, and the labeled samples are used for model performance evaluation.

#### 3.4.2. Parameter Settings

Similar to the case study of the TE chemical process, the optimal parameters for CTC process are determined as follows:(i)MWGPR: the width of the moving window is set to 43.(ii)JITGPR: the number of local modeling samples is set to 43, and the best similarity is cosine similarity;(iii)OSELM: prediction block is set to 1 to predict one value at a time, the number of hidden neurons is set to 38, and the number of initial training samples used in the initial phase is set to 43.(iv)OSVR: penalty parameter C is set to 24, tuning parameter for kernel function g is set to 0.02, and precision threshold p is set to 0.007.(v)PALM: the optimal parameters are the same as TE industrial process. $b_{1}$, $b_{2}$, $c_{1}$, and $c_{2}$ are set to 0.02, 0.01, 0.01, and 0.01, respectively;(vi)OSEGPR: the number of online validation sample sets is set to 43, the ensemble size is set to 5, and Manhattan distance similarity is chosen.(vii)SS-OSEGPR: the number of online validation sample sets is set to 43, the ensemble size is set to 5, the confidence threshold for selecting pseudo-labels is set to 0.1, and Manhattan distance similarity is chosen.(viii)ODCSS_S_: clustering radius R is set to 4.9, minimum density threshold M is set to 12, and maximal ensemble size m is set to 2;(ix)ODCSS: clustering radius R is set to 4.8, minimum density threshold M is set to 14, controlling parameter α is set to 0.6, proportion of deleted data P is set to 0.9, confidence threshold $IR_{th}$ is set to 0.1, and maximal ensemble size m is set to 3.

#### 3.4.3. Prediction Results and Discussion

Table 7 compares the best prediction performance of different soft sensor methods on the CTC process. As can be seen, OSVR and OSELM achieve poor performance on substrate concentration prediction. JITGPR provides better performance than MWGPR and PALM performs better than other single-model methods. In comparison, the proposed ODCSS method still shows the lowest RMSE, MAE, MAPE, and the highest $R^{2}$, implying the best prediction performance. Overall, the performance improvement of the proposed ODCSS approach compared to MWGPR, JITGPR, OSELM, OSVR, PALM, OSEGPR, and SS-OSEGPR methods is 21.4%, 16.2%, 25.8%, 26.3%, 11.7%, 10.6%, and 9%, respectively, when using RMSE as the baseline. After adding the pseudo-labeled data, the proposed ODCSS shows a 7% performance improvement compared with ODCSS_S._ These results further confirm that ODCSS has significantly better prediction accuracy than traditional soft sensors for semi-supervised data streams.

Figure 8 shows the scatter plots of the prediction results from different methods. The closer the scatter points are to the diagonal line, the more accurate the prediction results are. All compared methods exhibit different degrees of deviation. Among them, the scatter points of OSELM (Figure 8c) and OSVR (Figure 8d) are far from the diagonal line throughout the prediction zone. The rest of the methods have a tendency to deviate from the diagonal line at different zones. In contrast, the proposed ODCSS method (Figure 8i) obtains the closest and most dense overall scatter points to the diagonal line, thus providing the best prediction performance.

In addition, the dynamic clustering process is also illustrated through three-dimensional variables obtained by PCA. As shown in Figure 9, the red triangular points are outliers and the remaining colors correspond to clusters. Although some of the data are mixed after visualization using PCA, it does not affect the understanding of the clustering process in this paper. As Figure 9b shows, when the 48th sample arrives, the size of data in the cluster and outliers increases over time. The 49th sample arrives and the outliers within the radius accumulate to the set threshold and a new cluster is formed, as shown by the pink points in Figure 9c. When the data in a cluster are accumulated to a certain extent, the obsolete samples in the cluster are deleted in order to improve operation efficiency and assure prediction accuracy, as shown in Figure 9d,e. Figure 9f–i shows the repetition of the above process.

Moreover, we further explore the influences of the parameters on the proposed ODCSS algorithm, as shown in Table 8. After fixing clustering radius R as 4.8, minimum density threshold M as 14, and maximum ensemble size m as 3, prediction performance of ODCSS under different combinations of the ratio of the deleted data $P\in \left[0.4,0.6,0.8\right]$, controlling parameter $\alpha \in \left[0.7,0.9\right]$, and confidence threshold $IR_{th}\in \left[0.05,\;0.2,0.15\right]$ are compared. These results show the strong stability and excellent accuracy of the proposed ODCSS method when using varying model parameters.

Similar to TE process, a Freidman test with a Finner post hoc method is conducted in order to further assess different soft sensor methods in the CTC fermentation process. For this purpose, the test set was divided in batch order and a total of 13 batches of RMSE values are obtained, and then a non-parametric Friedman test is performed. The statistical test results are also tabulated in Table 9. It can be readily observed that, similar to TE process, MWGPR, JITGPR, OSELM, and OSVR reject the $H_{0}$ hypothesis, which indicates that the prediction performance of ODCSS is remarkably different from these methods in CTC process. In comparison, PALM, OSEGPR, SS-OSEGPR, and ODCSS_S_ accept the $H_{0}$ hypothesis, which reveals that there is no significant difference between ODCSS and the other compared methods in terms of the overall prediction performance.

To further explore the prediction capability of the different methods on local prediction stages, the RMSE values from different soft sensor methods on the 13 test batches are presented in Table 10. As can be seen from the table, the proposed ODCSS method shows poor prediction performance in the first two batches of prediction and then provides the best prediction accuracy in most later batches. Similar to TE process, these results once again confirm the strong online learning capability of ODCSS. However, we can also notice that ODCSS_S_ provides poor prediction RMSE for batch 11. Such a problem may be addressed by introducing other adaptation mechanisms to further enhance the online learning capability of ODCSS in accommodating complex data stream environments.

## 4. Conclusions

This paper presents an online-dynamic-clustering-based soft sensor (ODCSS) for industrial semi-supervised data streams. By applying online dynamic clustering to process data streams, ODCSS enables automatic generation and update and deletion of obsolete samples, thus realizing dynamic identification of process state. In addition, an adaptive switching prediction method combining online selective ensemble with JITL is used to effectively handle gradual and abrupt time-varying features, thus preventing model degradation. Moreover, to tackle the label scarcity issue, semi-supervised learning is introduced to obtain high-confidence pseudo-labeled samples online. The proposed ODCSS is a fully online soft sensor method that can effectively deal with nonlinearity, time variability, and shortage of labeled samples in industrial data streaming environments.

To verify the effectiveness and superiority of the proposed ODCSS method, two application cases are considered. Meanwhile, seven representative soft sensor methods and ODCSS_S_ (without pseudo-labeled samples) are compared with the proposed ODCSS. From the RMSE, MAE, MAPE, and $R^{2}$, it is evident that the proposed method outperforms the other compared methods in terms of all evaluation metrics. Especially, in TE process, with RMSE as a baseline, ODCSS improves prediction accuracy by 48.7% compared to OSELM, and introduction of semi-supervised learning improves prediction performance by 3.6% compared to ODCSS_S_. For the CTC fermentation process, although ODCSS does not show significant differences from some methods in terms of overall testing performance, the superiority of the proposed method becomes more and more obvious with accumulation of streaming data and advancement in online learning. Both the TE and CTC application results confirm that the proposed ODCSS method can well address time-variability, nonlinearity, and label scarcity problems and thus achieve high-precision real-time prediction of subsequent online arrived samples by using only very few labeled samples and adding high-confidence pseudo-labeled data.

Currently, there is still a lack of research on soft sensor modeling for data streams, and this study is only a preliminary attempt. There are still several issues requiring further attention. First, although the proposed algorithm has good prediction accuracy, computational burden of online modeling will inevitably increase with accumulation of process data streams. Thus, how to improve the efficiency of online modeling is also a major concern. Second, with evolution of process data streams, optimal model parameters also change, so it is appealing to adjust the model parameters adaptively. Third, the proposed method only considers identifying local features based on the spatial relationship between samples. For data streams, temporal relationships between samples are also worth noting. Fourth, as streaming data accumulate, mining the hidden features of streaming data using incremental deep learning is also an interesting research direction. These issues remain to be studied in our later work.

## Figures and Tables

**Figure 1 sensors-23-01520-f001:**
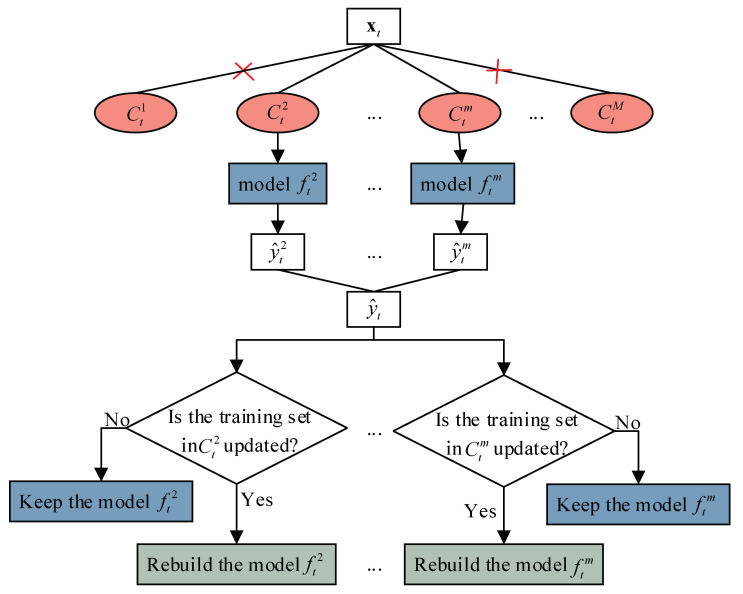
Schematic diagram of adaptive selective ensemble learning framework.

**Figure 2 sensors-23-01520-f002:**
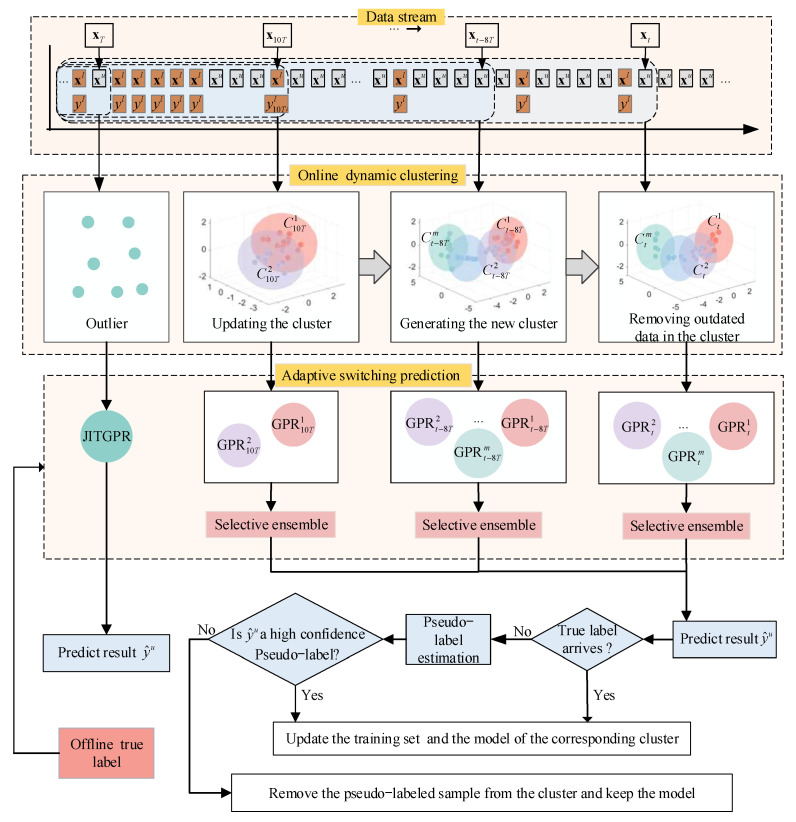
The overall framework of the ODCSS soft sensor method.

**Figure 3 sensors-23-01520-f003:**
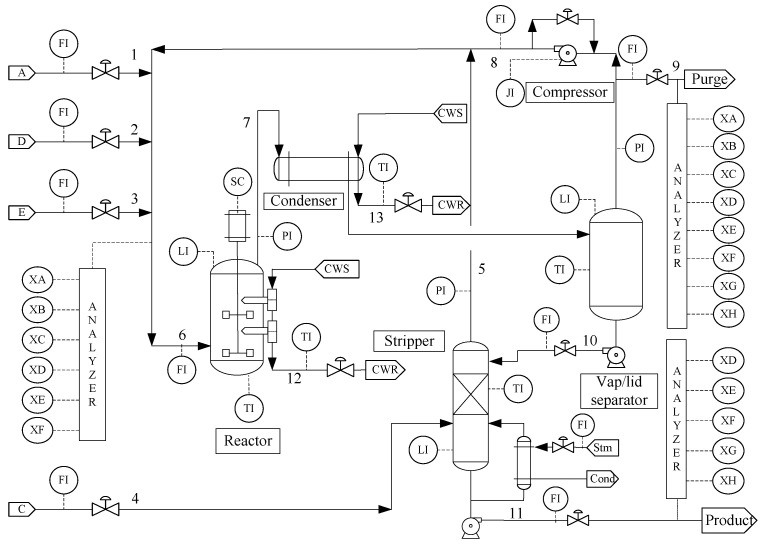
Flowchart of TE chemical process.

**Figure 4 sensors-23-01520-f004:**
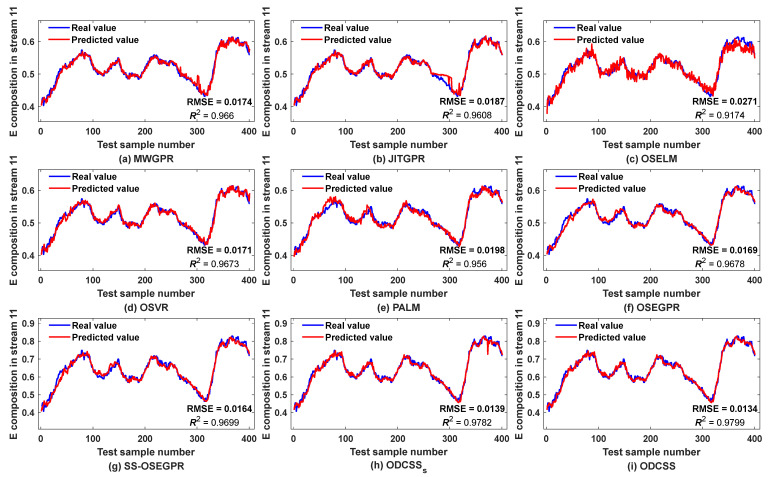
Prediction trends of E component in stream 11 by different soft sensor methods.

**Figure 5 sensors-23-01520-f005:**
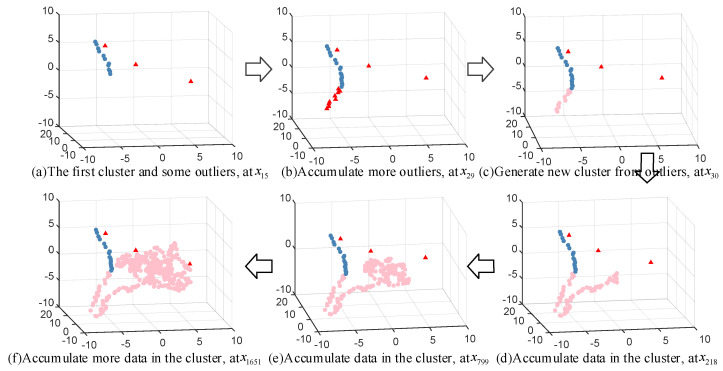
Illustrations of online dynamic clustering for TE process.

**Figure 6 sensors-23-01520-f006:**
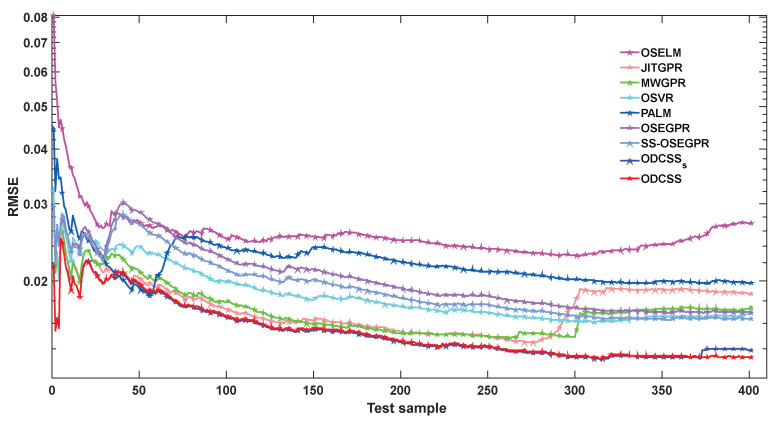
Comparison of RMSE trends of different soft sensor methods.

**Figure 7 sensors-23-01520-f007:**
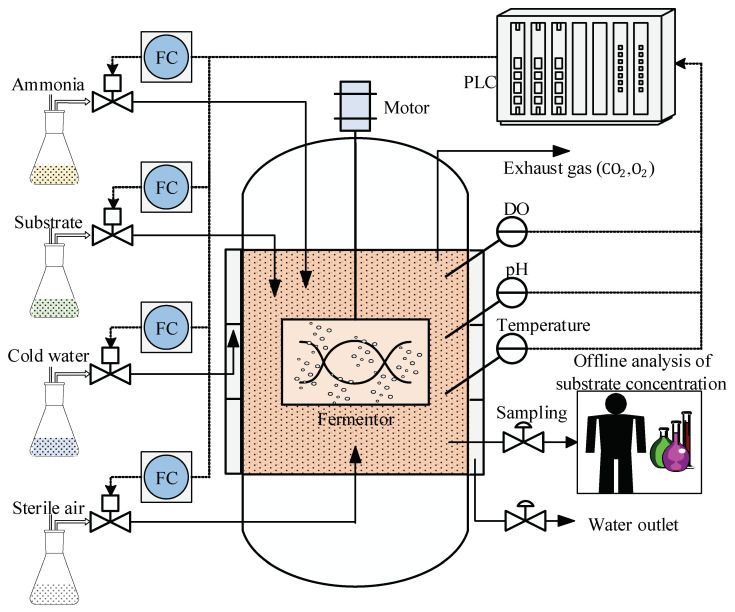
Flowchart of industrial fed-batch CTC fermentation process.

**Figure 8 sensors-23-01520-f008:**
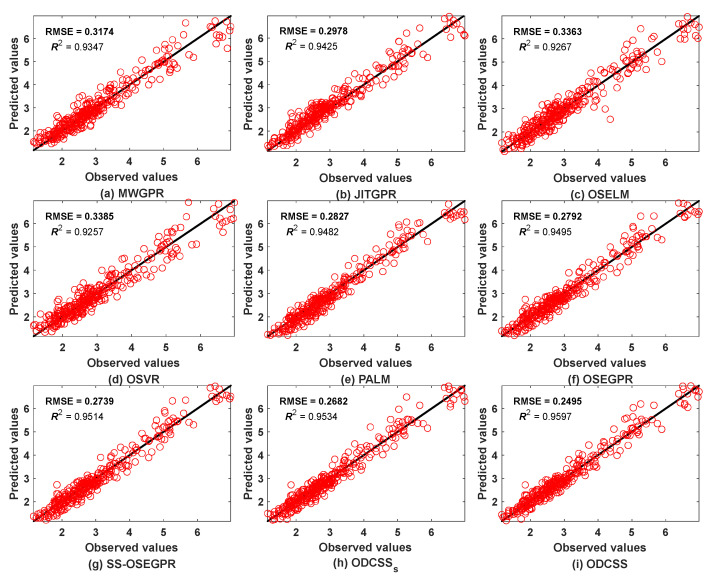
Scatter plots of substrate concentration predictions by different soft sensor methods.

**Figure 9 sensors-23-01520-f009:**
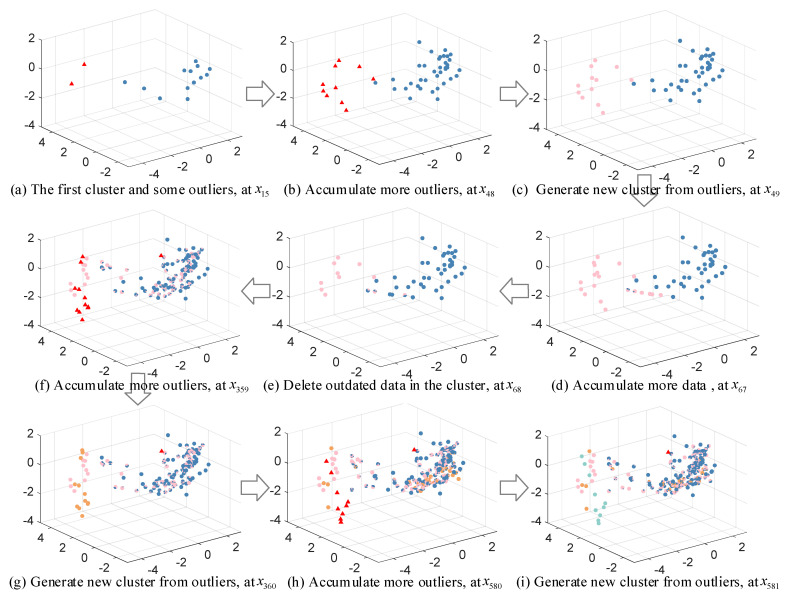
Illustrations of online dynamic clustering for CTC process.

**Table 1 sensors-23-01520-t001:** Input variables for soft sensor development in TE chemical process.

No.	Variable Description	No.	Variable Description
1	Time	17	Stripper pressure
2	A feed (stream 1)	18	Stripper underflow
3	D feed (stream 2)	19	Stripper temperature
4	E feed (stream 3)	20	Stripper steam flow
5	A and C feed rate	21	Compressor work
6	Recycle flow rate	22	Reactor coolant temperature
7	Reactor feed rate	23	Separator coolant temperature
8	Reactor pressure	24	D feed flow (stream 2)
9	Reactor level	25	E feed flow (stream 3)
10	Reactor temperature	26	A feed flow (stream 1)
11	Purge rate	27	A and C feed flow (stream 4)
12	Product separator temperature	28	Purge valve (stream 9)
13	Product separator level	29	Separator pot liquid flow (stream 10)
14	Product separator pressure	30	Stripper liquid product flow (stream 11)
15	Product separator underflow	31	Reactor cooling water flow
16	Stripper level	32	Condenser cooling water flow

**Table 2 sensors-23-01520-t002:** Comparison of different soft sensor methods for prediction of TE composition in stream 11.

Method	RMSE	MAE	MAPE (%)	$R^{2}\;$
MWGPR	0.0174	0.0130	2.1017	0.9660
JITGPR	0.0187	0.0138	2.2709	0.9608
OSELM	0.0271	0.0212	3.3412	0.9174
OSVR	0.0171	0.0137	2.2020	0.9673
PALM	0.0198	0.0158	2.4966	0.9560
OSEGPR	0.0169	0.0131	2.1106	0.9678
SS-OSEGRP	0.0164	0.0128	2.0488	0.9699
ODCSS_S_	0.0139	0.0109	1.7397	0.9782
ODCSS	0.0134	0.0107	1.7168	0.9799

**Table 3 sensors-23-01520-t003:** Performance comparison of the proposed method under different parameters for prediction of E composition in stream 11.

No.	R	M	RMSE	$R^{2}$
1	8	10	0.0135	0.9797
2	9	10	0.0134	0.9799
3	10	10	0.0135	0.9796
4	11	10	0.0134	0.9798
5	12	10	0.0138	0.9787
6	13	10	0.0134	0.9797
7	8	12	0.0135	0.9795
8	9	12	0.0134	0.9798
9	10	12	0.0135	0.9797
10	11	12	0.0137	0.9790
11	12	12	0.0136	0.9792
12	13	12	0.0137	0.9791
13	8	14	0.0136	0.9792
14	9	14	0.0134	0.9798
15	10	14	0.0135	0.9795
16	11	14	0.0134	0.9798
17	12	14	0.0136	0.9793
18	13	14	0.0135	0.9795

**Table 4 sensors-23-01520-t004:** Statistical test results for the RMSE differences between ODCSS and different compared soft sensor methods in TE process.

Method	Statistical Test		
	Statistic	*p*-Value	Result
MWGPR	2.77390	0.00995	Reject $H_{0}$
JITGPR	2.11058	0.03907	Reject $H_{0}$
OSELM	5.84932	0	Reject $H_{0}$
OSVR	3.61814	0.00067	Reject $H_{0}$
PALM	4.46237	0.00002	Reject $H_{0}$
OSEGPR	2.71360	0.00997	Reject $H_{0}$
SS-OSEGRP	2.23118	0.03288	Reject $H_{0}$
ODCSS_S_	0.12060	0.90400	Accept $H_{0}$
ODCSS (the control method)	-	-	-

**Table 5 sensors-23-01520-t005:** The prediction RMSE values of different soft sensor methods on the test subsets in TE process.

No.	Method									ODCSS Rank
	MWGPR	JITGPR	OSELM	OSVR	PALM	OSEGPR	SS- OSEGRP	ODCSS_s_	ODCSS	
1	0.0219	0.0219	0.0269	0.0244	0.0233	0.0237	0.0232	0.0206	0.0205	1
2	0.0195	0.0180	0.0269	0.0229	0.0135	0.0325	0.0297	0.0179	0.0183	4
3	0.0140	0.0148	0.0231	0.0159	0.0344	0.0132	0.0135	0.0132	0.0135	3
4	0.0142	0.0117	0.0243	0.0142	0.0171	0.0130	0.0119	0.0124	0.0122	3
5	0.0098	0.0118	0.0257	0.0131	0.0198	0.0162	0.0160	0.0116	0.0115	2
6	0.0130	0.0171	0.0237	0.0174	0.0287	0.0204	0.0191	0.0148	0.0150	3
7	0.0133	0.0120	0.0265	0.0151	0.0162	0.0124	0.0124	0.0120	0.0120	2
8	0.0119	0.0111	0.0203	0.0115	0.0121	0.0095	0.0095	0.0089	0.0089	2
9	0.0148	0.0148	0.0164	0.0161	0.0168	0.0153	0.0149	0.0133	0.0133	1
10	0.0123	0.0098	0.0190	0.0111	0.0179	0.0135	0.0134	0.0101	0.0101	2
11	0.0168	0.0165	0.0171	0.0118	0.0127	0.0116	0.0120	0.0107	0.0107	1
12	0.0304	0.0431	0.0242	0.0135	0.0148	0.0111	0.0109	0.0092	0.0092	1
13	0.0206	0.0213	0.0337	0.0209	0.0170	0.0155	0.0154	0.0150	0.0150	1
14	0.0195	0.0196	0.0335	0.0179	0.0218	0.0160	0.0166	0.0128	0.0128	1
15	0.0177	0.0130	0.0432	0.0207	0.0181	0.0162	0.0156	0.0183	0.0127	1
Mean	0.0166	0.0171	0.0256	0.0164	0.0189	0.0160	0.0156	0.0134	0.0131	1

**Table 6 sensors-23-01520-t006:** Input variables of soft sensor models for industrial fed-batch CTC fermentation process.

No.	Variable Description
1	Cultivation time (min)
2	Temperature (°C)
3	PH (−)
4	Dissolved oxygen concentration (%)
5	Air stream rate ($m^{3}/h$)
6	Volume of air consumption ($m^{3}$)
7	Substrate feed rate (L/h)
8	Volume of substrate consumption (L)
9	Volume of ammonia consumption (L)

**Table 7 sensors-23-01520-t007:** Comparison of different soft sensor methods for prediction of substrate concentration in CTC fermentation process.

Method	RMSE	MAE	MAPE (%)	$R^{2}\;$
MWGPR	0.3174	0.2405	8.5315	0.9347
JITGPR	0.2978	0.2266	8.0920	0.9425
OSELM	0.3363	0.2544	9.0366	0.9276
OSVR	0.3385	0.2621	9.2811	0.9257
PALM	0.2827	0.2178	7.9556	0.9482
OSEGPR	0.2792	0.2135	7.6421	0.9495
SS-OSEGRP	0.2739	0.2082	7.4001	0.9514
ODCSS_S_	0.2682	0.2005	7.1162	0.9534
ODCSS	0.2495	0.1841	6.4624	0.9597

**Table 8 sensors-23-01520-t008:** Performance comparison of the proposed method under different parameters for substrate concentration prediction.

No.	P	α	$IR_{th}$	RMSE	$R^{2}$
1	0.4	0.7	0.05	0.2558	0.9576
2	0.6	0.7	0.05	0.2573	0.9571
3	0.8	0.7	0.05	0.2687	0.9532
4	0.4	0.9	0.05	0.2540	0.9582
5	0.6	0.9	0.05	0.2551	0.9578
6	0.8	0.9	0.05	0.2535	0.9584
7	0.4	0.7	0.1	0.2533	0.9584
8	0.6	0.7	0.1	0.2543	0.9581
9	0.8	0.7	0.1	0.2596	0.9563
10	0.4	0.9	0.1	0.2591	0.9565
11	0.6	0.9	0.1	0.2495	0.9597
12	0.8	0.9	0.1	0.2531	0.9585
13	0.4	0.7	0.15	0.2568	0.9573
14	0.6	0.7	0.15	0.2535	0.9584
15	0.8	0.7	0.15	0.2659	0.9542
16	0.4	0.9	0.15	0.2512	0.9591
17	0.6	0.9	0.15	0.2545	0.9580
18	0.8	0.9	0.15	0.2608	0.9559

**Table 9 sensors-23-01520-t009:** Statistical test results for the RMSE differences on the testing subsets between ODCSS and other compared soft sensor methods in CTC fermentation process.

Method	Statistical Test		
	Statistic	*p*-Value	Result
MWGPR	3.23875	0.00270	Reject $H_{0}$
JITGPR	2.85010	0.00785	Reject $H_{0}$
OSELM	3.36830	0.00227	Reject $H_{0}$
OSVR	3.95128	0.00035	Reject $H_{0}$
PALM	1.55460	0.13400	Accept $H_{0}$
OSEGPR	2.07280	0.05674	Accept $H_{0}$
SS-OSEGRP	1.61938	0.13338	Accept $H_{0}$
ODCSS_S_	1.16595	0.24363	Accept $H_{0}$
ODCSS (the control method)	-	-	-

**Table 10 sensors-23-01520-t010:** The prediction RMSE values of different soft sensor methods on the test batches in CTC fermentation process.

No.	Method									ODCSS Rank
	MWGPR	JITGPR	OSELM	OSVR	PALM	OSEGPR	SS- OSEGRP	ODCSS_S_	ODCSS	
1	0.3169	0.3148	0.5521	0.3223	0.2872	0.3724	0.3441	0.2726	0.3321	6
2	0.4614	0.2720	0.3908	0.4051	0.1984	0.1823	0.1985	0.2075	0.3117	6
3	0.2173	0.2940	0.3561	0.3170	0.2018	0.2167	0.2160	0.2013	0.1880	1
4	0.2987	0.2978	0.2943	0.3106	0.2051	0.2330	0.2202	0.1875	0.1904	2
5	0.2359	0.2597	0.1869	0.2470	0.2033	0.1598	0.1459	0.1701	0.1506	2
6	0.3406	0.3530	0.2413	0.3909	0.2491	0.2723	0.2751	0.2332	0.2153	1
7	0.3155	0.2437	0.3635	0.3827	0.4303	0.3986	0.3810	0.3950	0.2232	1
8	0.3118	0.2122	0.2686	0.3649	0.2522	0.2684	0.2657	0.2576	0.2006	1
9	0.2677	0.3026	0.3300	0.2594	0.2221	0.2776	0.2755	0.3193	0.2530	2
10	0.3342	0.3045	0.3578	0.3528	0.3296	0.2947	0.2975	0.3166	0.2862	1
11	0.3704	0.2994	0.2662	0.3880	0.2949	0.2539	0.2530	0.2829	0.2968	6
12	0.3189	0.3775	0.3021	0.3616	0.2847	0.3134	0.3146	0.2858	0.2746	1
13	0.2582	0.2739	0.2897	0.2568	0.3902	0.2824	0.2818	0.2669	0.2302	1
Mean	0.3113	0.2927	0.3230	0.3353	0.2730	0.2712	0.2668	0.2613	0.2425	1

## Data Availability

The data cannot be made public because of privacy restrictions.

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
