# Peer review of "Online-Dynamic-Clustering-Based Soft Sensor for Industrial Semi-Supervised Data Streams"

_sensors, 2023, doi:10.3390/s23031520_

Round 1

Reviewer 1 Report

This paper presentes an online dynamic clustering based soft sensor (ODCSS) for industrial semi-supervised data streams. The method achieves automatic generation and update of the clusters and samples deletion through online dynamic clustering, thus enabling online dynamic identification of process states. Meanwhile, selective ensemble learning and just-in-time learning (JITL) are employed through an adaptive switching prediction strategy, which allows dealing with gradual and abrupt changes in process characteristics and thus alleviates model performance degradation caused by concept drift. In addition, semi-supervised learning is introduced to exploit the information of unlabeled samples and obtain high-confidence pseudo-labeled samples to expand the labeled training set. 

Some comments:

1) The paper alternates the description of the work with the description of the (standard) techniques adopted. This makes it difficult to distinct the actual innovation proposed by this work from the (standard) methods. Maybe the description of the standard techniques can be moved to an appendix, in order to highlight the original contributions.

2) The validation by a domain expert of the correct of the results provided by  provided by online dynamic clustering based soft sensor (ODCSS) is missing in the discussion section. Maybe, the optimal model parameters can be chosen by a domain expert to improve the quality of clustering and semi-supervised data streams.

In conclusion, this manuscript addresses the specific problem. The paper is well written but the lacks a meaningful domain expert evaluation and validation and an extensive comparison between the results obtained and those proposed in the literature are missing. No methodological approach is provided, but only the application of some standard ML algorithms and models provided in the literature without any particular innovation/improvement. For this type of submission, this approach is far from sufficient.

Reviewer 2 Report

Dear all, 

the paper is well defined and presented. I have nothing to add before the publication. 

Best regards,

Author Response

Thank you for your time and patience.

Reviewer 3 Report

The abstract should mention significance of your study, like why this topic is important, method used why etc.

In the introduction, what key theoretical perspectives and empirical findings in the main literature have already informed the problem formulation? What major, unaddressed puzzle, controversy, or paradox does this research address? 

What are the key issues the present study has addressed?

Why are authors focused on Online dynamic clustering?

What are the practical implications of your research? 

CONCLUSIONS is too short. Add more explanation. 

What are the limitations of the present work?

Check all the text for: mathematical description, typographical errors, and grammatical mistake

Round 2

Reviewer 1 Report

The authors correctly answered all the questions, thus enriching their work correctly and comprehensively. This work can be accepted in the present form

Reviewer 3 Report

.